# MMC Transformer: Multiscale Multigrid Comparator Transformer for Few-Shot Video Segmentation

## Abstract

Learning to compare support and query feature sets for few-shot image and video understanding has been shown to be a powerful approach. Typically, methods limit feature comparisons to a single feature layer and thus ignore potentially valuable information. In particular, comparators that operate with early network layer features support precise localization, but lack sufficient semantic abstraction. At the other extreme, operating with deeper layer features provide richer descriptors, but sacrifice localization. In this paper, we address this scale selection challenge with a meta-learned Multiscale Multigrid Comparator (MMC) transformer that combines information across scales. The multiscale, multigrid operations encompassed by our architecture provide bidirectional information transfer between deep and shallow features (*i.e.* coarse-to-fine and fine-to-coarse). Thus, the overall comparisons among query and support features benefit from both rich semantics and precise localization. Additionally, we present a novel multiscale memory learning in the decoder within a meta-learning framework. This augmented memory preserves the detailed feature maps during the information exchange across scales and reduces confusion among the background and novel class. To demonstrate the efficacy of our approach, we consider two related tasks, few-shot video object and actor/action segmentation. Empirically, our model outperforms state-of-the-art approaches on both tasks.

## 1   Introduction

Guided by a few labelled examples (*i.e.* the support set), few-shot learning is focused on improving the generalization ability of models to novel classes unseen during the initial training to classify the query images. In this paper, our focus is on metric learning for few-shot video segmentation, where methods learn to compare features between the support and query sets (*i.e.* learning comparators). Previous dense prediction work has documented that pixel-to-pixel comparisons between the query and support sets better capture fine details compared to working with global average pooled representations [19, 21]. A key question arises: Which features should be compared? Limiting comparisons between the support and query sets at the finest scales (*i.e.* shallow network layers) capture only primitive semantics (*e.g.* local orientation) and thus are error prone, while the coarser scales (*i.e.* deeper network layers) capture abstract semantics but sacrifice detailed information that support precise localization.

To address this scale selection challenge, we present a novel Multiscale Multigrid Comparator (MMC) transformer that takes as input a set of correlation tensors that encompass comparisons between support-query features at multiple abstraction levels. Our transformer incorporates multigrid processing [3, 2] that allows bidirectional information to be exchanged across scales (*i.e.* coarse-to-fine and fine-to-coarse). Critically, this multiscale information exchange reduces the impact of erroneous correlations at the finer scales by incorporating feedback from coarser scales and allows finer scale

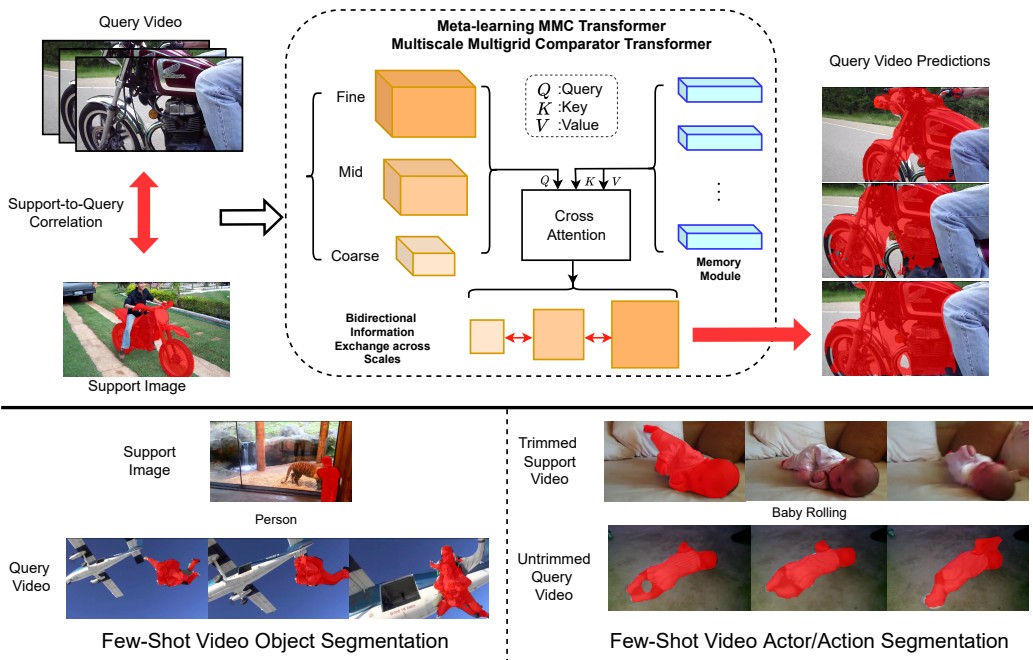

Figure 1: Overview of our Multiscale Multigrid Comparator (MMC) transformer. **Top:** We take as input correlations between backbone features at different scales for both support and query sets. We meta-learn our MMC transformer that encompasses multiscale cross attention between the spatial/spatiotemporal features at different levels and a memory module that helps separate the background and novel class. Our MMC transformer allows bidirectional information exchange across scales following the multigrid formulation between fine-mid-coarse scale features in the multiscale cross attention blocks. Additionally, our design preserves the spatiotemporal dimension during the information exchange across scales through multiscale memory learning (*i.e.* $K$ and $V$) that help separates the background and novel class. **Bottom:** We demonstrate our approach on two video tasks that require dense predictions: few-shot video object and action segmentation. The support set groundtruth and query predictions are highlighted in red.

information to feedback to the coarser scales, allowing more detailed information to modulate the coarse-grained information. A key enabler of MMC is allowing the multiscale processing within the transformer decoder to preserve the spatiotemporal dimension, rather than pooling the information into a compact vector [7, 6]. Finally, to address the issue of confusions between the novel class and background we use a novel multiscale memory learning module. As two illustrative video tasks, we instantiate our model for few-shot video object (FSVOS) and actor/action segmentation. Fig. 1 provides an overview of our overall approach. Since our paper explores both few-shot learning and transformers, to reduce ambiguities between the term *query* used in both, we use the term *target query* when referring to its usage in few-shot throughout the rest of the paper.

**Few-shot learning.** Metric learning (*i.e.* learning to compare) is a widely adopted approach in few-shot classification (*e.g.* [20, 22, 1, 24]), segmentation (*e.g.* [26, 21, 19]), video object segmentation [5] and action localization in video [29]. Multiscale processing often is not exploited in this paradigm [5, 29], even though it has the potential to enrich the representations over which support-to-target query comparators operate. While work in few-shot segmentation has considered multiscale processing in comparators [19], the model can be confused by erroneous correlations, especially at the finest scales. In our work, we investigate meta-learning a multiscale transformer with a memory module that better separates the background from the novel class by allowing bidirectional information exchange across scales.

**Multiscale transformers.** Incorporating multiscale processing in transformers is an emerging topic (*e.g.* action recognition [10, 15] and panoptic segmentation [7]). Previous work has mainly explored multiscale information on the encoder side [10, 15, 27], which is not sufficient when computing

dense predictions. In recent work [7, 6], a multiscale transformer decoder was proposed, but its design exchanged information across scales on the compressed learnable queries. In our approach we preserve the spatiotemporal dimension in the multiscale processing to enable a multigrid formulation. Thus, information exchange across scales is on the detailed feature maps rather than a compressed representation. We also operate on correlation tensors rather than directly on features maps to induce a stronger bias towards learning a comparator. Closely related to our work is the task of finding dense correspondences between images, where both global and local information is combined using transformers [8]; however, they have no notion of memory or meta-learning. In contrast, our work explores multiscale memory learning in the transformer decoder within a meta-learning framework. There has been previous works on memory augmented transformers [30, 18]; however, our approach is the first to explore multiscale memory learning in the transformer decoder. This novelty is crucial to preserve the spatiotemporal dimension during the cross scale information exchange and to better separate the background and novel class. Additionally, we establish connections to classical multiscale processing methods [3], which has not been explored in the context of transformers.

**Multigrid methods.** Multigrid methods [3, 2] were initially developed to accurately solve large systems of partial differential equations in a computationally efficient manner and with reduced residual error. They operate on multiple discretization levels, where interactions among fine and coarse grids occur when deriving the approximate solution. The V-cycle correction scheme [3] is one form of a recursive solution that allows bidirectional information flow across the different scales to ensure smooth solutions with low error as well as efficient computation. This approach inspires our multigrid formulation that reduces the effect of erroneous correlations across the scales via bidirectional information exchange. While previous work has integrated multigrid-like operations in convolutional architectures [14], we are the first to explore multigrid processing within multiscale transformers. Additionally, our formulation is cast within a meta-learning framework that targets few-shot learning tasks associated with dense predictions in videos.

**Contributions.** In this paper, we present a novel comparator for few-shot learning tasks associated with dense predictions in videos. Our main contributions are threefold: (i) We present the first attempt to meta-learn a multiscale comparator between the support and target query sets in few-shot video dense prediction tasks. (ii) Our comparator encompasses a multiscale, multigrid transformer decoder that operates on correlation tensors between the support and target query set features with bidirectional multiscale information exchange. (iii) We present multiscale memory learning in the transformer decoder within a meta-learning framework that operates on top of the correlation feature pyramid to better separate the novel class from the background. We demonstrate our MMC transformer on two few-shot video tasks, few-shot video object and action segmentation, where our method outperforms the state of the art on both tasks. Our code will be publicly released upon acceptance.

## 2   Multiscale multigrid transformer comparator

In this section, we detail our multiscale comparator design. Inspired by classical multigrid methods [3], we develop a formulation that allows bidirectional information exchange across scales. Additionally, to allow for information exchange across scales using detailed feature maps we perform cross attention with a learnable memory module and preserve the spatiotemporal dimension of the feature maps within our decoder. The proposed memory module helps to distinguish the novel class from the background and enhances support-to-target query correlation features.

### 2.1   Multiscale comparator transformer

Since we are operating on multiple levels of feature abstractions and resolutions, we use the subscript $_p$ to denote the features from scale level $p \in 1, 2, \ldots, P$, which are extracted from late (coarse), $p = 1$, intermediate or early (fine), $p = P$, stages. The input features for level $p$ after flattening are $Z_p \in \mathbb{R}^{TH_pW_p \times C_p}$, where $H_p, W_p$ are spatial dimensions, $T$ is the clip length and $C_p$ are the channels for the corresponding scale. The input features are constructed on the support-query correlation tensors, as detailed in Sec. 3. We further project it with a $1 \times 1$ convolutional layer, to reduce the dimensionality for a memory efficient solution, and end up with $\bar{Z}_p \in \mathbb{R}^{TH_pW_p \times D}$.

We start by defining (multihead) attention [25] as,

$$\mathcal{A}_h(X^q, X^k, X^v) = \text{Softmax}\left(\frac{X^qW^q(X^kW^k)^\top}{\sqrt{D}}\right)X^vW^v, \tag{1}$$

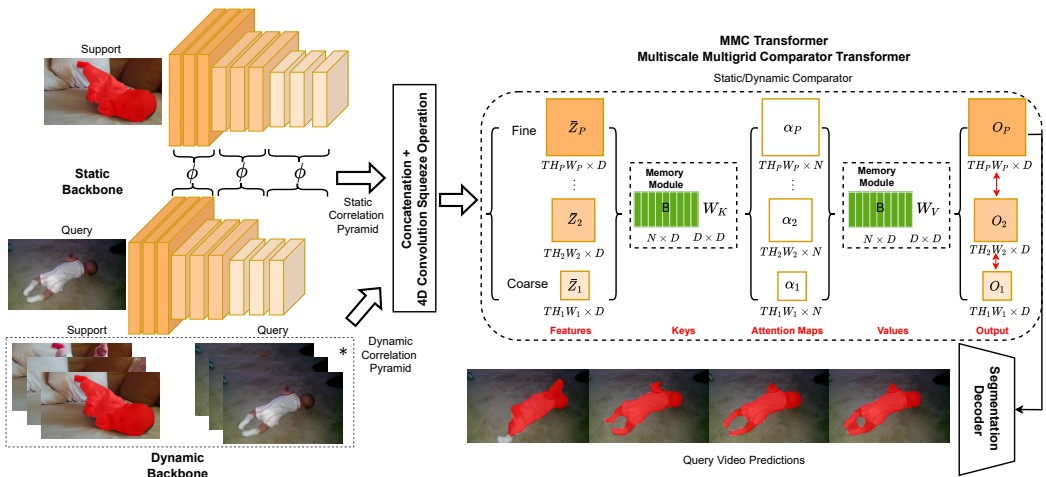

Figure 2: Overview of our architecture for few-shot video object and action segmentation. Features from a convolutional backbone that capture information conveyed from single static images (*a.k.a.* static backbone) are extracted. It can optionally (∗) be combined with features from a convolutional backbone that capture dynamics conveyed from a set of frames (*a.k.a.* dynamic backbone). 4D correlation tensors are computed among spatial/spatiotemporal features from support and target query sets using $\phi$. Subsequently, the tensors go through a 4D convolution that yields 2D feature maps for memory efficiency. A multigrid multiscale comparator transformer is used on the input pyramid, $\{\bar{Z}_p\}_{p=1}^P$. Cross attention among the feature pyramid and the learnable memory, $B$, is performed to generate attention maps, $\{\alpha_p\}_{p=1}^P$. The cross attention module encompasses key, $W_k$, and value, $W_v$, weight matrices. The attention maps are used to re-weight the memory, $B$, in each spatiotemporal position to enhance query features and separate the background from the novel class in the output, $\{O_p\}_{p=1}^P$, Eq. 2. Information is transferred bidirectionally between scales (denoted by red arrows) and the final output at the finest scale is used as input to the segmentation decoder.

109 where $\mathcal{A}_h$ represents the attention per head, $h$, and the full multihead attention is, $\mathcal{A}$, that corresponds
110 to the concatenation of each head's output. The inputs $X^q, X^k, X^v$ represent the query, key, value,
111 resp., and $W^q, W^k, W^v \in \mathbb{R}^{D \times D}$ are the query, key and value weight matrices, resp., for $D$ feature
112 dimensions. We use fixed spatiotemporal positional embeddings, $E_p^s \in \mathbb{R}^{T H_p W_p \times D}$, corresponding
113 to every scale level, $p$, and learnable scale embeddings, $E_p^l \in \mathbb{R}^{1 \times D}$, following [7]. We repeat the
114 scale embedding at all spatiotemporal positions, $T, H_p, W_p$, resulting in $\hat{E}_p^l \in \mathbb{R}^{T H_p W_p \times D}$.

115 Importantly, we seek a formulation that preserves the spatiotemporal dimension during the information
116 exchange across scales in the multiscale transformer decoder and cross attention. This ensures the
117 multiscale processing and information transfer can capture detailed information in the feature maps
118 rather than a compressed representation. Therefore, in contrast to previous work that instantiate
119 a set of learnable queries [7], we meta-learn a memory module that has $D$ dimensional vectors,
120 $B \in \mathbb{R}^{N \times D}$, with $N$ memory entries that are shared across all decoding layers and scales. This
121 multiscale memory learning allows the per-scale decoded output to preserve the spatiotemporal
122 dimension, unlike [7]. We perform cross attention per resolution and feature abstraction level, $p$,
123 where we instantiate the multihead attention, (1), as

$$O_p = \mathcal{A}(\bar{Z}_p + \hat{E}_p^l + E_p^s, B + E_b, B), \tag{2}$$

124 where $E_b \in \mathbb{R}^{N \times D}$ are learnable memory positional embeddings. Thus, we perform multiscale
125 processing while maintaining the spatiotemporal dimension for the output, $O_p \in \mathbb{R}^{T H_p W_p \times D}$, as
126 illustrated in the MMC transformer block of Fig. 2. This mode of operation ensures cross-scale
127 communcation with the detailed feature maps and maintains the spatiotemporal dimension output
128 from our decoder. For every scale level $p$, applying $\mathcal{A}$ will learn to attend among the different set of
129 learnable memory features based on their relevance to the support-target query correlation features. It
130 then aggregates the learned memory based on these attention maps to better separate the novel class
131 and the background. Since we want to allow for information exchange across scales, we use

$$\bar{Z}_{p\prime} = \bar{Z}_{p\prime} + I_p^{p\prime} O_p, \tag{3}$$

where $I_p^{p\prime}$ performs bilinear interpolation to match the size from level $p\prime$. The cross attention operations, (2), are performed consecutively on all $P$ levels and are repeated $N_l$ times, with $N_l$ a hyperparameter denoting the number of decoder layers for each level. The final output from our multiscale comparator is $O_P$ that will be used later for the final segmentation prediction. Since our design does not collapse the spatiotemporal dimension, the output, $O_p$, for every level, $p$, maintains detailed information necessary for the final segmentation task, unlike previous work used in a non-meta-learning framework for panoptic segmentation [6].

Typically, previous work has focused on multiscale processing in the transformer decoder with multiscale query learning that contextualizes a set of learnable features, $Q \in \mathbb{R}^{N \times D}$ [7, 6]. The multiscale query learning output can be seen as,

$$O_p^{\text{query}} = \mathcal{A}(Q + E_b, \bar{Z}_p + \hat{E}_p^l + E_p^s, \bar{Z}_p), \tag{4}$$

where $O_p^{\text{query}} \in \mathbb{R}^{N \times D}$ are a set of compressed learnable queries that are exchanged across scales. This formulation uses learnable features, $Q$, as queries and the multiscale feature maps, $\bar{Z}_p$, as keys and values resulting in outputs per scale of dimension, $N \times D$. In contrast, our formulation of multiscale processing with a memory module uses the learnable features, $B$, as keys and values; hence, the queries are the detailed feature maps, $\bar{Z}_p$, which yields per-scale output of dimension, $T H_p W_p \times D$. Thus, the detailed feature maps necessary for segmentation are lost during the communication across scales in the former, but are preserved in ours. We empirically validate this distinction in Sec. 4.2.

## 2.2 Multiscale multigrid attention (MMA)

Now that we have presented a means to leverage multiscale processing in the transformer comparator, we further explore the different forms of information transfer across different scales. Communication across scales can be conducted in (i) coarse-to-fine processing that is performed in a *stacked* manner for multiple layers, as shown in Fig. 3 (left) or (ii) a *multigrid* formulation that allows for bidirectional information transfer between the coarse and fine scales, as shown in Fig. 3 (right). Bidirectional exchange ensures that the coarse-grained smoothed correlation features with high level semantics can modulate information in the fine-grained ones, while allowing fine-grained detailed information to affect the coarse-grained.

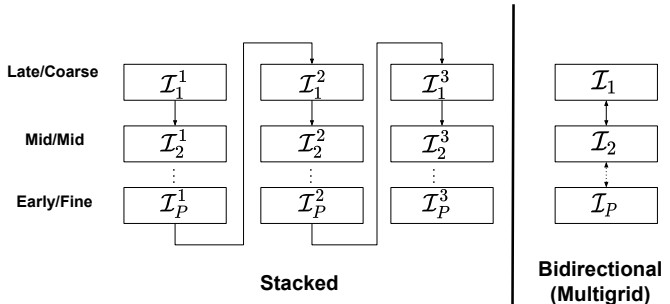

Figure 3: Two variants of information exchange across scales with different feature abstraction/resolution levels: *stacked vs. multigrid*. Feature abstraction levels indicate early-to-late stage features and different resolution levels indicate coarse-to-fine levels. $\mathcal{I}_p$ denotes multihead attention, (1), for level $p$ followed by bilinear interpolation, (3). In the *stacked* variant, $\mathcal{I}_p^j$ is for the $j^{th}$ iteration.

Inspired by classical multigrid methods, we present a multiscale, multigrid attention module that allows for bidirectional information transfer across different scales, as shown in Fig. 3 (right), similar to the V-cycle correction scheme [3]. In our case, since the multiscale, multigrid attention operates on correlation tensors from different scales, it ensures smooth solutions through bidirectional information exchange between coarse and fine scales. It thereby avoids erroneous correlations that might exist at the different scales, where the fine scale can exhibit erroneous correlations as it only captures low level semantics and the coarse scale can exhibit noisy correlations from being a subsampled signal. Unlike classical multigrid methods our multiscale input exhibits both different levels of resolution as well as different feature abstraction levels (*i.e.* early, mid and late stage features), instead of solely using different sampling rates on the same information as classical multigrid. Another perspective

that motivates the coarse-to-fine and fine-to-coarse communication is inspired from earlier work that has shown convolutional layers mainly act as a band pass filtering [12], which indicates that features from every layer capture a certain set of frequency components. Thus, the bidirectional information transfer can enrich the features through exchanging information across the different set of frequencies captured at each layer.

Consider the *stacked* multihead attention, which takes as input multiscale pyramids. It is composed of a series of multihead cross attention (2), followed by merging the two consecutive levels and using bilinear interpolation to match the scale, (3). The combined aforementioned operations are denoted as, $\mathcal{I}_p$ for level $p \in \{1, 2, \ldots, P\}$. Thus, we formulate *stacked* multihead attention (SMA) in a coarse-to-fine processing per iteration as

$$\text{SMA} = \mathcal{I}_1^j \circ \mathcal{I}_2^j \circ \cdots \circ \mathcal{I}_P^j, \tag{5}$$

where $\circ$ denotes function composition and $\mathcal{I}_p^j$ corresponds to the $j^{th}$ iteration as the operations in the *stacked* multihead attention are repeated $N_l$ times. In comparison, *multigrid* multihead attention (MMA) processing is defined as

$$\text{MMA} = \mathcal{I}_1 \circ \mathcal{I}_2 \circ \cdots \circ \mathcal{I}_P \circ \mathcal{I}_{P-1} \circ \cdots \circ \mathcal{I}_1 \circ \mathcal{I}_2 \circ \cdots \circ \mathcal{I}_P. \tag{6}$$

It is seen that the multigrid approach, (6), encompasses bidirectional information exchange across scales similar to classical recursive methods [3], whereas the stacked approach is strictly coarse-to-fine. Sec. 4.2 provides empirical support for the superiority of the *multigrid* formulation.

# 3    Learning scheme

In this section, we summarize the few-shot video setup, and our scheme for meta-learning the multiscale comparator with multigrid, multiscale attention for an improved few-shot video tasks. Then we describe two case studies for few-shot video object and actor/action segmentation.

**Few-shot setup.**  We formulate the few-shot video object or actor/action segmentation task as follows [5, 29]. Let $\mathcal{D}_{train}$ and $\mathcal{D}_{test}$ be training and testing data, resp. For every dataset, we split the $C$ categories into $O$ folds, each fold will have $\frac{C}{O}$ novel categories, $\mathcal{C}_{test}$, and $C - \frac{C}{O}$ as base classes, $\mathcal{C}_{train}$. Both the training and test classes do not intersect, $\mathcal{C}_{train} \cap \mathcal{C}_{test} = \emptyset$. In the meta-training phase, we sample $N_e$ tasks from the corresponding dataset with support and target query set pairs $\{\mathcal{S}_i, \mathcal{Q}_i\}_{i=1}^{N_e}$ for classes in $\mathcal{C}_{train}$. Similarly in meta-testing we sample support and target query sets but for classes in $\mathcal{C}_{test}$. The target query set contains video frames $\mathcal{Q} = \{X_t^{(q)}\}_{t=1}^{N_v}$, where $N_v$ is the number of frames and superscript $\_^{(q)}$ denotes the target query set. In the case of video object segmentation, the support set in a one-way $K$-shot task has $K$ image-label pairs $\mathcal{S} = \{X_k^{(s)}, M_k^{(s)}\}_{k=1}^K$ for a class to be separated from the background. The superscript $\_^{(s)}$ denotes support set and $M_k^{(s)}$ is a binary segmentation mask for the class considered. The image-label pairs $X_k \in \mathbb{R}^{H \times W \times 3}$ and $M_k \in \mathbb{R}^{H \times W}$, with $H \times W$ spatial dimensions. In the case of video actor/action segmentation the one-way $K$-shot task has $K$ trimmed video-label pairs $\mathcal{S} = \{X_k^{(s)}, M_k^{(s)}\}_{k=1}^K$. The binary segmentation mask $M_k^{(s)}$ is for one frame in the trimmed video for an actor/action class. Thus, the video-label pairs are $X_k \in \mathbb{R}^{T \times H \times W \times 3}$ and $M_k \in \mathbb{R}^{H \times W}$.

**Meta-learning a multiscale comparator.**    We start with introducing an overview of the full architecture of our multiscale comparator, as shown in Fig. 2. We initially assume a one-shot setting, then discuss the K-shot extension later. We use a pretrained convolutional backbone with fixed weights, $F$, that are not updated during the meta-training process, to compute the support and target query set features. The features for the one-shot support and target query sets are extracted for layer, $l$, as, $f_l^{(s)} = F_l(X^{(s)}), f_l^{(q)} = F_l(X^{(q)})$, resp., for the set of $L$ layers. Let $\phi(\cdot, \cdot)$ denote the (hyper)correlation encompassing the comparisons between its arguments, both of which are tensors, and $\bigoplus$ be concatenation on the channel dimension for $m$ consecutive layers with the same spatial dimensions. Then, we define a 4D hypercorrelation tensors pyramid with $P$ levels as $H_p = \bigoplus_l^{l+m} \phi(f_l^{(s)}, f_l^{(q)})$, *cf.* [19]. We use the hypercorrelation squeeze network [19], $D_{\text{hypercorr}}$, which represents one form of performing efficient 4D convolution and generates a 2D feature pyramid that is further flattened, $\{Z_p\}_{p=1}^P = D_{\text{hypercorr}}(\{H_p\}_{p=1}^P)$.

Our multiscale comparator transformer uses the features extracted on different levels by performing cross attention to a learnable memory, $B$, to yield the final output feature maps, $O_P =$

232   $D_{\text{multiscale}}(\{Z_p\}_{p=1}^{P}, B)$. The cross attention re-weights the memory features to enhance the query
233   feature maps and separate the background from the novel class based on their correlation to the
234   support set. Attention and feature aggregation subsequently is computed in a pixel-wise manner
235   across all levels to produce the final output features, $O_P$. Our multiscale multigrid transformer
236   decoder enriches the features and allows bidirectional exchange of information across scales. The
237   output features, $O_P$, from the multiscale comparator are used in a segmentation decoder, $D_{\text{seg}}$, to
238   compute the final predictions $\hat{M} = D_{\text{seg}}(O_P)$. The predictions, $\hat{M}$, are for $N_c$ classes that include
239   the background class. The hypercorrelation squeeze network, $D_{\text{hypercorr}}$, our multiscale comparator,
240   $D_{\text{multiscale}}$, and segmentation decoder, $D_{\text{seg}}$, are meta-trained with a simple binary cross entropy that
241   encourages the model to segment the class of interest guided by the support set. During few-shot
242   inference when operating with K-shot support set, we follow the setup from [19] which infers the
243   target query prediction with every example in the support set separately, sums all predictions and
244   divides by the maximum score.

245   **Static/Dynamic comparator transformer.** In this section, we introduce two related tasks (*i.e.*
246   few-shot video object segmentation and few-shot video actor/action segmentation) and show how
247   our multiscale multigrid comparator operates on both. The term *static* factor indicates information
248   learned from a single frame (*e.g.* texture and colour), while the *dynamic* factor indicates information
249   extracted from a consecutive set of frames (*e.g.* motion). We meta-learn a static comparator in the case
250   of few-shot video object segmentation, while in the case of few-shot video actor/action segmentation
251   we find it beneficial to meta-learn both a static/dynamic comparator, as shown in Fig. 2. We describe
252   each setting in turn next.

253   In the case of few-shot video object segmentation, which is a simpler task, the goal is to segment the
254   novel class per target query frame in the video. Since dynamics might not have a significant effect on
255   identifying the semantic categories (*e.g.* person moving or standing is class person) we design a static
256   comparator. We use per-frame features extracted from a 2D backbone (*e.g.* ResNets [13]). The support
257   set in this setup is a set of single images that can already describe the semantic category. Thus, the fea-
258   tures extracted for support and target query are $f_l^{(s)} \in \mathbb{R}^{H_l \times W_l \times C_l}$ and $f_l^{(q)} \in \mathbb{R}^{T \times H_l \times W_l \times C_l}$, resp.
259   The corresponding hypercorrelation pyramid is computed as the set of correlation tensors for each
260   target query frame in the video and the support set features, $\{H_p \in \mathbb{R}^{T \times H_p \times W_p \times H_p \times W_p \times C_p}\}_{p=1}^{P}$.
261   Consequently, the extracted features from the hypercorrelation squeeze network that is applied in-
262   dividually per frame and flattened, $\{Z_p \in \mathbb{R}^{T H_p W_p \times C_p}\}_{p=1}^{P}$, are used in our MMC transformer to
263   generate enhanced query features, $O_P \in \mathbb{R}^{T H_P W_P \times D}$. Finally, the segmentation predictions from
264   the decoder is given by $\hat{M} \in \mathbb{R}^{T \times H \times W \times N_c}$.

265   In the case of few-shot actor/action segmentation, the dynamic factor is important in identifying the
266   action while the static factor delineates the object/actor boundaries. Correspondingly, we meta-learn
267   a static/dynamic comparator that fuses the information from both. Toward this end we use a 3D
268   backbone (*e.g.* X3D [11]) to extract spatiotemporal features that we refer to as dynamic features. In
269   complement, we use a 2D backbone to extract the first frame features from the current input clip, we
270   refer to these as static features. We use superscript $.^{(dy)}, .^{(st)}$ to denote the corresponding dynamic
271   and static tensors, resp. Additionally, the support sets are trimmed videos instead of a single image.
272   Thus, the dynamic features extracted are given as $f_l^{(s)}, f_l^{(q)} \in \mathbb{R}^{T \times H_l \times W_l \times C_l}$. The corresponding
273   dynamic hypercorrelation pyramid is computed after averaging along the temporal dimension in each
274   layer, then computing the correlation tensors, $\{H_p^{(dy)} \in \mathbb{R}^{H_p \times W_p \times H_p \times W_p \times C_p^{(dy)}}\}_{p=1}^{P}$. Similarly,
275   the static hypercorrelation pyramid is built on top of the features extracted for the first frame in
276   the current clip as $\{H_p^{(st)} \in \mathbb{R}^{H_p \times W_p \times H_p \times W_p \times C_p^{(st)}}\}_{p=1}^{P}$. Then, we combine the correlations
277   from both static and dynamic features to yield our final pyramid, $\{H_p = H_p^{(dy)} \oplus H_p^{(st)}, H_p \in$
278   $\mathbb{R}^{H_p \times W_p \times H_p \times W_p \times C_p^{(st)} + C_p^{(dy)}}\}_{p=1}^{P}$. The extracted features from the hypercorrelation squeeze are
279   flattened as $\{Z_p \in \mathbb{R}^{H_p W_p \times C_p}\}_{p=1}^{P}$ and are used as inputs by our MMC transformer to generate the
280   final features, $O_P \in \mathbb{R}^{H_P W_P \times D}$. Finally, the segmentation prediction for an input clip from the
281   decoder is given by $\hat{M} \in \mathbb{R}^{H \times W \times N_c}$. We use a temporal sliding window over the untrimmed target
282   query video and generate clips that are used to predict the segmentation.

# 4 Experimental results

## 4.1 Experiment design

**Datasets and evaluation protocol.** We evaluate on two standard benchmarks, YouTube-VIS FS-VOS [5] and Common A2D [29], to facilitate comparison to the state of the art in few-shot video object and action segmentation. We follow their standard evaluation protocol and describe the details in the supplement.

**Implementation details.** For few-shot video object segmentation we follow the same architectural choices as state-of-the-art approaches [5] to facilitate comparison, where we build on a ResNet-50 [13] backbone pretrained on ImageNet [9]. For few-shot actor/action segmentation, we use both ResNet-50 pretrained on ImageNet [9] and X3D [11] pretrained on Kinetics [4] following [29] for the static and dynamic backbones, resp. In our MMC transformer, the number of decoder layers per scale is set to $N_l = 3$ and the number of entries in our learned memory is $N = 20$. We meta-train our model and baseline [19] on the base classes for a given fold using cross entropy, with 50 epochs on YouTube-VIS, while we use 70 epochs for A2D. We use the same hyperparameters for both our approach and the baseline, where we use AdamW [17] with a learning rate of $1 \times 10^{-3}$ and weight decay of $1 \times 10^{-4}$. Random rotations and flipping data augmentation is used for both. Additional implementation details are provided in the supplement.

## 4.2 Ablation study

We start an ablation study on our two main contributions: multigrid (*i.e.* bidirectional) information exchange across scales and multiscale memory learning which preserves the spatiotemporal dimension during multiscale processing. In Table 1, we compare four variants: (i) our baseline without a transformer decoder [19], (ii) the multiscale transformer with learnable queries that pools the spatiotemporal dimension following (4), which we call *Query*, (iii) our multiscale transformer that preserves the spatiotemporal dimensions, (2), and follows a stacked information flow across scales as shown in Fig. 3 (left), which we call *Stacked* and (iv) our MMC transformer with bidirectional information flow as in Fig. 3 (right), which we call *Multigrid*. We can see that across all folds the mIoU for *Query* is lower than any of the variants (*i.e. Stacked* and *Multigrid*) and is even lower than the baseline on average. It shows our approach of multiscale memory learning to preserve the spatiotemporal dimension in the multiscale transformer comparator yields more accurate segmentations. Additionally, we ablate both forms of information flow across scales, *Stacked*, following (5) and *Multigrid* following (6). Here, it is seen that the bidirectional information flow in the multigrid approach improves over the stacked coarse-to-fine across three of the four folds. It is also seen that our multigrid approach improves over the baseline [19] on average and especially on the first two folds. We hypothesize that the failures in the last two folds were due to under-segmentation exhibited by our comparator that made the model more restrictive on what is considered part of the novel class, qualitative examples presented in the supplement. For example, in split two class *Hand*, although the target query video should have the entire arm segmented the support set mainly has the hand without the arm.

In Table 3, we compare our baseline [19], the improved baseline (*i.e.* baseline++) that combines correlation tensors from static and dynamic factors and our full approach MMC transformer with static and dynamic features on Common A2D. A greater improvement is seen with respect to the baseline than in object segmentation ablation, with up to 7% gain in the five-shot scenario. Additionally, these results demonstrate the flexibility of our multiscale comparator, as it can operate with any backbone network (*e.g.* ResNet-50 [13], X3D [11] or the combination of the two) and is able to operate beyond few-shot video object segmentation to actor/action segmentation.

## 4.3 Comparison to state-of-the-art approaches

We provide a comparison with existing approaches on YouTube-VIS FS-VOS in Table 2, where our method shows a notable gain of 4.4% with respect to the recently presented many-to-many attention comparator [5]. This result demonstrates that our multigrid, multiscale comparator helps separate the novel class with respect to the background better than previous approaches. Table 3 shows comparisons with the state of the art for few-shot video actor/action segmentation, with focus on the

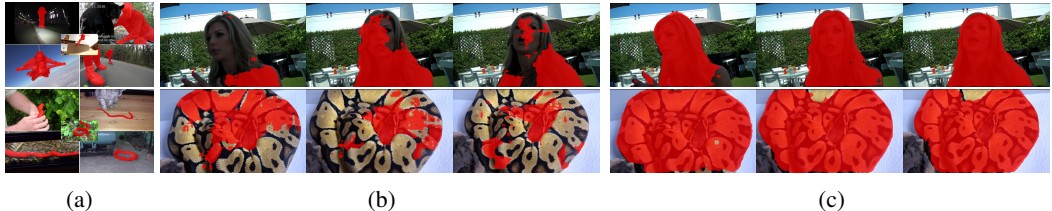

(a)               (b)                   (c)

Figure 4: Qualitative results showing better separation of background *vs.* novel class for our MMC transformer. (a) Five-shot support set. (b) Baseline [19]. (c) MMC transformer (ours). The support groundtruth and query predictions are marked in red. Video results are provided in the supplement.

| Method | mIoU | | | | |
|---|---|---|---|---|---|
| | 1 | 2 | 3 | 4 | Mean |
| Baseline | 49.5 | 69.5 | **63.8** | **65.2** | 62.0 |
| Query | 49.4 | 70.5 | 62.9 | 64.3 | 61.8 |
| Stacked | 50.7 | 69.8 | 63.2 | 64.4 | 62.0 |
| Multigrid | **51.5** | **70.6** | 63.0 | 64.6 | **62.4** |

Table 1: Ablation study on YouTube-VIS FS-VOS folds 1, 2, 3 and 4 using our MMC transformer with a five shot support set.

| Method | mIoU | | | | |
|---|---|---|---|---|---|
| | 1 | 2 | 3 | 4 | Mean |
| PMMs [28] | 32.9 | 61.1 | 56.8 | 55.9 | 51.7 |
| PFENet [23] | 37.8 | 64.4 | 56.3 | 56.4 | 53.7 |
| PPNet [16] | 45.5 | 63.8 | 60.4 | 58.9 | 57.1 |
| DANet w/o OL [5] | 41.5 | 64.8 | 61.3 | 61.4 | 57.2 |
| DANet [5] | 43.2 | 65.0 | 62.0 | 61.8 | 58.0 |
| MMC transformer | **51.5** | **70.6** | **63.0** | **64.6** | **62.4** |

Table 2: Comparison of mIoU for few-shot video object segmentation to the state of the art on YouTube-VIS folds 1, 2, 3 and 4 with a five-shot support set. DANet w/o OL indicates the variant without online learning.

| Method | Static | Dynamic | mIoU | |
|---|---|---|---|---|
| | | | 1-shot | 5-shot |
| Co-attention [21] | - | - | 43.3 | 44.8 |
| Single-scale Transformer [29] | - | - | 50.6 | 52.5 |
| Baseline | ✓ | ✗ | 18.1 | 21.9 |
| Baseline++ | ✓ | ✓ | 45.2 | 47.7 |
| MMC transformer (Ours) | ✓ | ✓ | **51.9** | **54.5** |

Table 3: Comparisons of mIoU for few-shot actor/action segmentation with respect to the state of the art on Common A2D and our baselines, with one-shot and five-shot support sets. Static/Dynamic indicates the use of the corresponding factor, where the static compares ResNet-50 [13] spatial features and the dynamic compares X3D [11] spatiotemporal features.

segmentation task [29, 21]. It is seen that our approach consistently outperforms the others in both the one-shot and five-shot scenarios on Common A2D.

Fig. 4 shows qualitative results on YouTube-VIS FS-VOS, where our method improves over the baseline with better separation of the novel class with respect to the background and better delineated boundaries. We provide Common A2D qualitative results in the supplement.

## 5   Discussions and Conclusion

We presented a novel MMC transformer that exchanges bidirectional information across scales for comparing between the support and query sets and reducing the impact from erroneous correlations. Our transformer decoder is designed to preserve the spatiotemporal dimension in our bidirectional multiscale processing, unlike previous methods. We meta-learn our multiscale comparator transformer along with a memory module that better separate the background from the novel class. We showcased the MMC transformer in two use cases, few-shot video object segmentation and actor/action segmentation. Our method outperforms the state of the art on both tasks. A limitation of our method for few-shot video object segmentation is its reliance on 2D backbone features, whereas in the video domain 3D spatiotemporal features may improve discriminability. We leave it for future work to explore spatiotemporal models as backbones and investigate FSVOS benchmarks that reflect the fact that certain semantic categories can exhibit different motion patterns (*e.g.* four legged *vs.* two legged mammals *vs.* reptile motion).

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
