# OpenReview forum: "MMC Transformer: Multiscale Multigrid Comparator Transformer for Few-Shot Video Segmentation"
_NeurIPS.cc/2022/Conference — NeurIPS 2022 Submitted_

### Official Review · Reviewer_ZBpu · 2022-07-09

**Rating:** 3
**Confidence:** 3
**Soundness:** 2 fair
**Presentation:** 1 poor
**Contribution:** 2 fair

**Summary:**

This paper presents a multi-scale transformer as a comparator in few-shots video segmentation tasks. Compared with prior works that mostly use one feature resolution for comparison, they leverage multi-scale comparison to obtain both coarse and fine associations. A multi-grid mechanism is used for communication between different scales. The authors use a pretrained memory module to perform cross attention with the correlation volume. The resultant method outperforms current state-of-the-art methods in few-shot video object segmentation and few-shot actor segmentation.

**Questions:**

Is there any running time analysis? See also weaknesses.

**Limitations:**

The authors mention the use of 2D features as a limitation.

**Strengths And Weaknesses:**

Strengths:
- This paper is well-motivated. Having multi-scale correlation is a sensible solution to few-shots video segmentation problems. I also like the idea of keeping the spatial resolution in the transformer decoder which is again important to segmentation tasks.
- The proposed method achieves good results on benchmark datasets, which further support the previous point.


Weaknesses:
- I think the writing can be improved. The authors rely heavily on abstract notation and procedural manipulation of these symbols without a high-level picture (in section 3). Most symbols are also not semantic so readers have to go back and forth many times to recall the meaning of a symbol. Yet, section 2.2 is too high-level and does not connect with the rest of the paper. The exact mechanism of multigrid bidirectional communication is not well explained. Adding some figures or rewriting sections 2 and 3 might help. This obstructs my understanding of this paper so I am putting a low confidence.
- There should be multiple transformer decoder layers (L134). This is not shown in Figure 1 or 2 (like with a *n).
- A “memory” module should carry information across different timesteps, iterations or scales. Currently I find that the memory module is meta-learned and is not updated during inference time. If this is the case, it should not be called a memory module as it does not dynamically store information. It is more like a set of queries.
- The features for the support and target query are extracted from the same feature extractor (L223). This means the mask is not an input to the target query and the correlation volume does not carry mask information. I do not find this mask used as input in any other places. This does not make sense as the output should depend on the support mask.

---

> ### Author Response · Authors · 2022-08-02
> **R4 response**
>
> We thank the reviewer for the feedback and address concerns in the following. 1) "authors rely heavily on abstract notation and procedural manipulation of these symbols without a high-level picture":  The high-level picture was introduced in Sec.3 **meta-learning a multiscale comparator**, but we will add connections to Sec.2 for example L231-232 connects to Eqs.2,3,6 combined in $D_{multiscale}$. 2) "There should be multiple transformer decoder layers. This is not shown in Figure 1 or 2.": Decoder layers are shown in Figure 3, but are omitted in both Figures 1 and 2 for space constraints, we will clarify this point in the captions. 3) "I find that the memory module is metalearned and is not updated during inference time": In response, we propose to instead refer to  the memory module as *meta-learned read only memory*; what it is learned was discussed in Sec. 3.2, L64-67 Supplementary and further illustrated in the supplementary video. 4) "It is more like a set of queries.": We do not use the term queries, since the conventional method [7] learns a set of queries, which leads to a loss of the spatiotemporal dimension in the multiscale processing, and is contrast to our approach using keys and values, which does not suffer this disadvantage. 5) "I do not find this mask used as input in any other places": Regarding the support mask we do use it, as described in Section 3 few-shot setup, L209-217, and as shown in Figure 2, when masking the support set features before computing the correlation tensors. This operation is conducted for both our approach and the baseline (including baseline++). We will clarify this detail in the final version. 5) "run time analysis": Runtime analysis is conducted on a Titan-X GPU for ours. Originally, DANet [5] reported 20 seconds per video on a  2080Ti GPU, while ours we report 2.9 seconds per video on average. This is due to their use of online finetuning that can greatly increase the run time, while our approach performs direct inference without any online finetuning. Thus, our approach outperforms state-of-the-art in both mIoU and run time

---

> > ### Comment · Reviewer_ZBpu · 2022-08-08
> > **Response**
> >
> > Thank you for the response. I appreciate the clarification and run-time results.
> > The support mask input seems to be used in "masking the support set features". This information does not exist in either Figure 2 or L209-L217.
> >
> > I agree with reviewer HsqG that the connection to V-cycle is loose (Section 2.2 seems to be a far-stretch attempt to connect them), and with reviewer DSqR that the paper is hard to follow.
> >
> > Overall I think this paper has potential but requires rewriting. At its current stage, it is too confusing. Thus, I have updated my rating to Reject.

---

> > > ### Author Response · Authors · 2022-08-09
> > > **Response**
> > >
> > > We thank the reviewer for his comment we want to clarify further in L227-229 we do mention that we use the hypercorrelation squeeze network from[19] where they mention "masking the support set features". So it was implicitly mentioned by referring to this work that has it part of their method. Nonetheless, we agree that explicitly mentioning it in Sec.3 would be better to make our work self sufficient. However, we argue that our paper is already "well written" as referred to by Reviewer 3 and the method is "straightforward" as referred to by Reviewer 1. Thus, we ask the reviewer to look through his decision since he agrees the work "has potential", the small writing modification of adding the above line can easily be made in the final camera ready. If there are other concerns regarding the method we would be more than happy to clarify it while referring to the lines in the paper that describes it.
> > >
> > > Regarding the "loose connection to the V-cycle or W-cycle": our Relation to the V-cycle or W-cycle correction lies in the bidirectional exchange, hence is one of our contributions. With each information exchange in our approach we perform a multihead attention based decoding following the schedule (coarse-mid-fine-mid-coarse-mid-fine) scales. We do not perform direct error correction as the original approach, but the multihead attention does perform one way of enhancing the correlation features and avoiding erroneous correlations hence why we use the term "inspire" as in L173-175. We think  the core of the multigrid methods is the bidirectional connections that allows information exchange among the different scales and is not necessarily tied to a certain error correction scheme.

---

### Official Review · Reviewer_973M · 2022-07-10

**Rating:** 7
**Confidence:** 4
**Soundness:** 3 good
**Presentation:** 3 good
**Contribution:** 3 good

**Summary:**

The paper proposes a multi-scale multi-grid comparator transformer for Few-shot video segmentation that sims to overcome limitations of prior Few-Shot segmentation methods that limit feature comparisons to only a single feature layer thus ignoring information that might be valuable. Typically features in the early net work layers aid in precise localization and deeper layer features give higher level semantic information. The work aims to combine both coarse and fine-grained features via multi-scale and multi-grid operations to improve few shot segmentation. In addition, the work also introduces a novel multiscale memory learning in the decoder which helps preserve the details within feature maps across scales

Contributions of the work are stated as:
+ a meta-learning method for multiscale comparison of query and support features is introduced
+ this comparator is a multiscale, multigrid transformer decoder that allows bidirectional multiscale information flow
+ finally the work also proposes a multiscale memory learning module within the transformer decoder

**Questions:**

Questions:

Q) how does this method work on standard image few-shot segmentation tasks? since reference [19] in the paper was referred to, it would be nice to see results on COCO 20i, Pascal 5i and FSS 1000 datasets.

**Limitations:**

The work only talks about limitations in terms of 2D backbone features. But it would be important to examine failure cases and see why that is happening.

Additionally, the impact of pre-trained features can be examined to further understand how important such features are?



**Strengths And Weaknesses:**

Strengths
+ well written paper
+ tackles an important problem
+ proposes improvements and novel architectural additions that improves the prior state-of-the-art results by a large margin

Weakness
+ evaluation can be more rigorous
+ how does this method work on standard image few-shot segmentation tasks? since reference [19] in the paper was referred to, it would be nice to see results on COCO 20i, Pascal 5i and FSS 1000 datasets.



Reference [19] in paper: Juhong Min, Dahyun Kang, and Minsu Cho. Hypercorrelation squeeze for few-shot segmentation. In Proceedings of the IEEE International Conference on Computer Vision, pages 6941–6952, 2021

---

> ### Author Response · Authors · 2022-08-02
> **R3 response**
>
> We thank the reviewer for their review. "Evaluation can be more rigorous": Supplement, Sec. 3.3 provides an additional rigorous analysis by considering performance as a function of spatial frequency content of input imagery; in particular, restriction to lower frequency components. We outperform our baseline across different frequencies with up to 1% gain averaged over 4 folds and 5 runs per fold. Here, we additionally provide results of a high frequency analysis in Table 2 (rebuttal), with comparative performance between our approach and our baseline as the signal is progressively restricted to higher frequency components (e.g., edges and fine grain texture). While both results are degraded by such filtering, our's degrades at a slower rate. To compute the high frequency components we apply Fourier transform, then high pass filtering with the complement of butterworth low pass filter as $F=\frac{1}{1+\frac{D}{D_0}}$ with varying $D_0$ to control the cutoff frequency. Then use inverse Fourier transform and use the result as input to the models. As an additional analysis, we show comparative performance for various noise distributions in Table 3 (rebuttal). Again, the relative robustness of our approach is seen. Additionally, for a more rigorous evaluation we test our approach beyond the fewshot setup on automatic video object segmentation fully supervised training on DAVIS16 and YouTube-VOS as followed in MATNet [D]. We train a Video-Swin backbone with our proposed multiscale memory learning with bidirectional exchange across scales. We compare it to multiscale decoder that uses queries, drops the spatiotemporal dimension (Mask2Former [7]), and uses Video-Swin backbone. We refer to this as Query similar to the ablation in Table 1 (main). We evaluate on DAVIS16 [C], MoCA [A] and YouTubeObjects [B]. Our approach outperforms the baseline (Query) on the three datasets and the state-of-the-art of AVOS on two datasets as reported in Table 4 (rebuttal).
>
> A-Hala Lamdouar, Charig Yang, Weidi Xie, and Andrew Zisserman. Betrayed by motion: Camouflaged object discovery via motion segmentation. In Proceedings of the Asian Conference on Computer Vision, pages 488–503, 2020.
>
> B-Alessandro Prest, Christian Leistner, Javier Civera, Cordelia Schmid, and Vittorio Ferrari. Learning object class detectors from weakly annotated video. In Proceedings of the IEEE/CVF Conference on Computer Vision and Pattern Recognition, pages 3282–3289, 2012.
>
> C-Federico Perazzi, Jordi Pont-Tuset, Brian McWilliams, Luc Van Gool, Markus Gross, and Alexander Sorkine-Hornung. A benchmark dataset and evaluation methodology for video object segmentation. In Proceedings of the IEEE/CVF Conference on Computer Vision and Pattern Recognition, pages 724–732, 2016.
>
> D- Zhou, Tianfei, et al. "Motion-attentive transition for zero-shot video object segmentation." Proceedings of the AAAI Conference on Artificial Intelligence. Vol. 34. No. 07. 2020.
>
> Table 2: High Frequency Analysis. We evaluate on YouTube-VIS and report average over 4 folds and 5 runs per fold mIoU. D0 controls the cutoff frequency, see text.
> |  Method |  Original |  D0=10 | D0=20 |  D0=30 | D0=40  |  D0=50 |
> |---|---|---|---|---|---|---|
> |  Baseline |  62.0 |  19.1 | 16.2  | 13.7  | 12.9 | 10.9  |
> | Ours  |  **62.4** | **19.3**  | **17.7**  | **16.5**  | **14.8**  |  **13.9** |
>
> Table 3: Noise Analysis. We evaluate on YouTube-VIS and report average over 4 folds and 5 runs per fold mIoU. We use Gaussian additive noise with standard deviation of 0.1 and zero mean. Salt and pepper noise use 0.4% of the image size as the amount of introduced noise. Speckle noise we use a normal distribution multiplied by the image and added to the original.
> | Method  | Original  | Gauss.  |  SaltPepper |  Speckle |
> |---|---|---|---|---|
> | Baseline  | 62.0  | 53.1  | 60.3 | 27.7 |
> | Ours  | **62.4**  |  **54.2** | **61.0**  |  **29.2** |
>
>
> Table 4: Video Object Segmentation results in terms of mIoU.
> | Method  | DAVIS [C] | MoCA [A]  |  YTBOvjects [B] |
> |---|---|---|---|
> |  Query |  81.0  |  75.8 |  75.6 |
> |  Ours | **82.8**  | **78.4**  | **77.9**  |
>
> Finally, "how does this method work on standard image few-shot segmentation tasks?": We focus on video related tasks and how our multiscale memory learning provides a temporally consistent attention maps as discussed in Sec 3.2 Supplementary. Nonetheless we understand the concern on how we can go beyond our current task. As such we show the versatility of our approach by going beyond the fewshot task to the fully supervised automatic video object segmentation task as presented earlier.

---

> > ### Author Response · Authors · 2022-08-09
> > **Response**
> >
> > We wanted to clarify on the AVOS experiments that we are outperforming the state-of-the-art methods even in the fully supervised VOS task by evaluating with ResNet101 backbone to make it similar to SOA methods MATNet[A] and RTNet[B] shown in Table 5. Even with the weaker backbone of ResNet101 we show our method outperforms the SOA in both MoCA and YTBObjects without requiring explicit optical flow unlike both[A,B] that needs precomputed optical flow. This confirms how our method both outperforms the baseline and SOA for the fully supervised AVOS task and in the fewshot VOS. Additionally, it outperforms the baseline in the fewshot Actor/Action segmentation task. We promise to add these results to the supplementary in the final version.
> >
> > Table 5: Video Object Segmentation results in comparison to SOA in terms of mIoU.
> > | Method | Backbone  | DAVIS [C] | MoCA [A]  |  YTBObjects [B] |
> > |---|---|---|---|---|
> > |  MATNet[A] | ResNet101 | 82.4  | 64.2  | 69.0  |
> > |  RTNet[B] | ResNet101 | **84.3** | 60.7  | 71.0  |
> > |  Ours | ResNet101 | 77.9  | **66.4**  | **72.1**  |
> > |  Query | Video-Swin | 81.0  |  75.8 |  75.6 |
> > |  Ours | Video-Swin | 82.8  | **78.4**  | **77.9**  |
> >
> > [A] Zhou, Tianfei, et al. "Motion-attentive transition for zero-shot video object segmentation." Proceedings of the AAAI Conference on Artificial Intelligence. Vol. 34. No. 07. 2020.
> >
> > [B] Ren, Sucheng, et al. "Reciprocal transformations for unsupervised video object segmentation." Proceedings of the IEEE/CVF conference on computer vision and pattern recognition. 2021.

---

### Official Review · Reviewer_DSqR · 2022-07-11

**Rating:** 4
**Confidence:** 3
**Soundness:** 3 good
**Presentation:** 3 good
**Contribution:** 2 fair

**Summary:**

This paper proposes a multiscale multigrid comparator transformer to address the few-shot video segmentation task. Experiments on few-shot video object segmentation and actor/action segmentation show effectiveness.

**Questions:**

Please refer to the Weaknesses above for the details.

**Ethics Review Area:**

["I don’t know"]

**Limitations:**

Yes. There's no suggestions.

**Strengths And Weaknesses:**

Pos: A novel comparator for few-shot learning tasks associated with dense predictions in videos is presented.

Neg:
1. Please explain what is static backbone and dynamic backbone.
2. In Equation 1, what are X_q, X_k and X_v respectively?
3. Equation 3 seems to be only a Unidirectional cross-scale information exchange. Where does the bidirectional information exchange mentioned by the author reflect?
4. This paper is hard to follow.
5. Writing needs to be improved.

---

> ### Author Response · Authors · 2022-08-02
> **R2 response**
>
> We thank the reviewer for their feedback and respond to all concerns. 1) "Please explain what is static backbone and dynamic backbone": The static backbone is used to extract spatial features from a static image (i.e., per frame), L254-256, while the dynamic backbone is used to extract spatiotemporal features from the input clip, L265-268. In our implementation, the static backbone is a ResNet-50 and the dynamic backbone is X3D. We use correlation tensors from both backbones when we conduct few-shot actor/action segmentation since the static features help delineate object boundaries while the dynamic ones help capture spatiotemporal features recognizing a specific action. In the case of few-shot video object segmentation we use only the static backbone with ResNet-50. 2)" In Equation 1, what are $X_q$, $X_k$ and $X_v$ respectively?": These are inputs for query, key, values respectively in the multihead attention, L110. 3) "Equation 3 seems to be only a Unidirectional" Eq. 2 & 3 reflects what happens in one information exchange across scales regardless if it is bidirectional or unidirectional. The bidirectional exchange is captured by Eq.6, via the combined operation of Eq. 2 & 3 and denoted by $I_p$ for level $p$,
> L189-190. 4,5) "This paper is hard to follow", "Writing needs to be improved": No specifics are provided as to improvements needed. In contrast,  R1 states "method is straightforward" and R3 states "paper was well written". Overall, the reviewer gave us 3 for presentation; so, we hope that our responses, which will be included in the final version, have provided additional information needed to respond to your concerns.

---

### Official Review · Reviewer_HsqG · 2022-07-11

**Rating:** 4
**Confidence:** 3
**Soundness:** 3 good
**Presentation:** 2 fair
**Contribution:** 2 fair

**Summary:**

**Problem**: This work addresses the problem of identifying correspondences between a given query feature map and a "support"/reference feature map. The goal for the correspondences is to be able to perform few-shot object segmentation or actor segmentation in videos.

**Solution**: The paper proposes a novel architecture (MMC) that takes as input dense feature similarity tensors and outputs frame-wise segmentation maps for the videos. The input similarity tensors measure the similarity between the support image features and features of the query video frames, at multiple levels of a CNN feature pyramid. The proposed MMC model comprises of a transformer-based encoder that reasons about the similarities using information from all levels of the similarity tensor pyramid and a decoder to output the frame-wise segmentation masks.


**Questions:**

- See comment on "novelty" in weaknesses above.
- For the task of Few Shot Video Object Segmentation, it looks like the convention is to report two metrics: mIoU and contour accuracy (as in DANet [5]). Is there a reason the contour accuracy is not reported here?

**Limitations:**

Discussion on limitations of the presented model is very limited in the paper.
Discussion on societal impact is not covered in the paper.

**Strengths And Weaknesses:**

## Strengths

### Intuitive Architecture
The proposed architecture is fairly straightforward to follow. The model involves a transformer architecture that uses learned key/value features to compute the attention and output features. These output features are pooled across the feature pyramid using a bidirectional pooling method. Finally the pooled features are passed to a decoder model to output the segmentation maps.

### Results
The comparison to state-of-the-art approaches demonstrates that the presented model can produce improved segmentation maps. The quantitative results show improved mIoU values across the two tasks: object segmentation and actor segmentation.

---

## Weaknesses

### Novelty
While the results are impressive, in the current form of the text, it is difficult to recognize the novelty of the work. This is partly because the novelty or importance of some of the design choices seems overstated in the text.
- The idea of reasoning across multiple feature levels of a CNN/similarity tensors is not novel (as pointed out by the authors too). This is the key idea of the "Multiscale Multigrid Attention". However, the specific implementation of this idea could be unique/novel. But it is not clear whether this is the case here. Section 2.2 seems to be inflating the contribution here. It is not clear why the stacked idea is presented since it is not the obvious/conventional method of performing feature pyramid pooling (to my best knowledge). And in addition to that, the stacked method actually performs almost equally well (in Table 1). Furthermore, the relationship to V-cycle correction is extremely loose. These explanations make it a bit harder to discern what the contribution is here.
- One of the contributions is the idea of using a memory module. The memory module is essentially learned keys (and values) which are shared across different levels. This idea seems novel, but as a core contribution this has not been investigated enough experimentally. For example, is it necessary to share this across the different levels? Do the same features need to be used as keys and values?
- The text seems to be using the term meta-learning (and meta-training, meta-testing) a lot. Is this one of the contributions? It is not clear why this is referred to as meta-learning. As far as I understand, meta-learning is the idea of optimizing through gradient descent update steps (or other optimization processes) to facilitate few-shot transfer learning - possibly introduced here first:
>"Finn, C., Abbeel, P., & Levine, S. (2017, July). Model-agnostic meta-learning for fast adaptation of deep networks. In International conference on machine learning (pp. 1126-1135). PMLR."

---

> ### Author Response · Authors · 2022-08-02
> **R1 response**
>
> We thank R1 for their feedback. We have tried to address the concerns on the novelty above and here we clarify remaining concerns. (1) “Conventional feature pyramid pooling” We highlight that our baseline [19] is considered the conventional version since it follows the  feature pyramid network design. However, it operates on correlation tensors instead of feature maps. In Table 3 (main) we outperform the Baseline and Baseline ++, which is an improved version of it, with up to 5% gain. Additionally, Table 1 (main) shows improvement when averaged over the 5 runs and over 4 folds. Moreover, Supplement, Figure 3 shows improvements of up to 1% average over all runs and folds compared to our baseline in a frequency-based analysis. (2) “Table 1 Stacked vs Multigrid results are equally well” We note that both  use multiscale memory learning, which is one of our main contributions, while a stacked version but without our memory learning was ablated under Query in Table 1 (main). In this case comparing the Multigrid to the Query in Table 1 (main) we consistently improve over all folds with up to 2% gain in fold 1. Notably, we are the first to investigate the different forms of information exchange, which is why we consider both the stacked and multigrid versions; alternatively, when we compare to the conventional multiscale processing we refer to Baseline and Baseline++ results. 3) "Multiscale Memory module seems novel but has not been investigated enough experimentally” We refer to Sections 3.1 and 3.2 in the Supplement, which ablates the number of memory entries and visualize the attention maps produced by the multiscale memory module with different exchanges across scales. Additionally, as suggested by the reviewer, we have conducted an ablation on the keys and values being meta-learned separately and find the mIoU on Fold 1 (separate: 50.1% vs shared: 51.5%), which makes the shared better. 4) "relationship to V-cycle correction is extremely loose" Relation to the V-cycle or W-cycle correction lies in the bidirectional exchange, hence is one of our contributions. With each information exchange in our approach we perform a multihead attention based decoding following the schedule (coarse-mid-fine-mid-coarse-mid-fine) scales. 5) “Contour Accuracy” We have evaluated using this metric and additionally evaluated the runtime and summarize the final results in Table 1 (rebuttal).
>
> Table 1: Few-shot VOS comparison to state-of-the-art with additional metrics (runtime and Contour Accuracy). We evaluate on YouTube-VIS 4 folds. OL: indicates methods that use online learning which performs backbone finetuning during the few-shot inference. Our method without any backbone finetuning improves the mIoU while running in a computationally efficient manner. ContAcc: Contour Accuracy. Runtime is measured in seconds per video.
> | Method  | OL  | mIoU-Fold1  | mIoU-Fold2  | mIoU-Fold3  | mIoU-Fold4  | Mean  | Runtime  | ContAcc-Fold1  | ContAcc-Fold2 | ContAcc-Fold3  | ContAcc-Fold4  |  Mean |
> |---|---|---|---|---|---|---|---|---|---|---|---|---|
> | DANet [5]  | Yes  |  43.2 | 65.0 | 62.0  |  61.8 | 58.0 | 20 |   42.3 |  **62.6**  |**60.6**|  **60.0**  | **56.3** |
> |  MMC transformers (ours) | No  | **51.5** | **70.6**  | **63.0**  | **64.6** | **62.4**  |  **2.9** | **44.6**  |  59.5  | 53.3  | 56.1  |  53.2 |
>
> Runtime analysis is conducted on a Titan-X GPU for ours. Originally, DANet [5] reported 20 seconds per video on a 2080Ti GPU. This is due to their use of online finetuning that can greatly increase the run time, while our approach performs direct inference without any online finetuning. Thus, our approach outperforms state-of-the-art in both mIoU and run time, while in the contour accuracy we outperform the state-of-the-art in Fold 0 but are less on the rest of the folds. Note that the contour accuracy evaluates the F-score on the boundary solely of the predicted segmentation using morphological operations, where we found our output to suffer in certain folds (2,3,4). We will add these results to the final version. 6) “why this is meta-learning” In general, meta-learning is learning to learn, where we simulate the few-shot inference during the training through sampling support/query sets in what is referred to as episodic training. Some of these methods focus on the model initialization scheme, as the reviewer referred to (Finn et. al.,ICML,2017), while others including this paper focus on learning to compare [19]. We will add this clarification to the introduction in the final version.  7) “Societal impact not covered” Supplement, Sec. 6 discusses this aspect, which was not included in the main paper for space constraints.

---

### Author Response · Authors · 2022-08-02
**Strengths + Novelty**

We thank all the reviewers for their efforts and highlight some of their positive comments, R1:"results are impressive", "Intuitive architecture", R2:"A novel comparator for few-shot learning", R3:"a novel multiscale memory learning in the decoder", "well-written paper", "proposes improvements and novel architectural additions that improves the prior state-of-the-art results by a large margin", R4:"The paper is well motivated. I like the idea of keeping the spatial resolution in the transformer decoder which is again important to segmentation tasks".

In the following, we start by addressing R1's concern on novelty, since we think this point is especially crucial to clarify.  Unfortunately, R1 did not refer to specific work that lessons our novelty; in contrast, R2 and R3 both commend our novelty (see quotes above) and R4 does not question it. Overall, our claims to novelty have not been refuted, which we repeat here (see L83-92 in the submission):
1) Meta-learning of a multiscale comparator for few-shot video segmentation. Earlier work was restricted to single scale and did not operate on correlation tensors [5,29].
2) Multiscale memory learning with meta-learned memory entries that produce temporally consistent attention maps and help separate background from the novel class. Prior work did not follow a meta-learning scheme, used feature maps not correlation tensors and used queries instead of keys and values, which lead to the loss of the spatiotemporal dimension [7].
3) We are the first to study different forms of information exchange between scales in multiscale transformers. We show the bidirectional information exchange across scales, inspired by multigrid methods outperforms information exchange schemes used in previous work.

These advances have potential to benefit a multitude of future efforts that are beyond the few-shot setting and thereby impact the general machine learning community.

---

### Meta-Review · Area_Chair_Da4y · 2022-08-27

**Recommendation:** Reject
**Confidence:** Certain

**Metareview:**

The paper develops a multigrid variant of the transformer architecture and applies it to video segmentation tasks.  After the author response and discussion phase, one reviewer recommends accept, but three of the four reviewers lean towards rejecting the paper.  In discussion, these three reviewers all acknowledged having read the author rebuttal and chose not to improve their scores.  The common concerns voiced across these reviews center on questionable novelty, clarity of explanation, and incremental experimental impact.  The Area Chair has also taken a detailed look at the paper, reviews, and author responses, and agrees with the concerns raised by these three reviewers.

Reviewer HsqG notes that "the idea of reasoning across multiple feature levels of a CNN/similarity tensors is not novel (as pointed out by the authors too)" and "Section 2.2 seems to be inflating the contribution here."  Section 2.2 and the author response highlights "bidirectional information exchange across scales, inspired by multigrid methods" as a key contribution.  However, reference [14] (Ke et al., Multigrid neural architectures, CVPR'17) explores exactly this idea: bidirectional information flow across scales, within a CNN architecture and even utilizes the same "multigrid" terminology.  From the point of neural network architecture design, the current paper's novelty appears limited to adapting previously established ideas to transformers.  So as not to appear to overclaim, the paper needs a broader discussion of the relationship of the proposed design to [14] as well as other prior work spanning multiscale, multiresolution, and feature pyramid architectures.

Reviewer concerns over experiments include marginal gains over ablated variants of the system (Table 1), and mixed results in comparison to DANet provided in the author response.  Overall, the response left unresolved questions over of contribution novelty, presentation clarity, and practical impact.

**Award:**

No

---

### Decision · Program_Chairs · 2022-09-14

Reject